# Test and Evaluation of Heart Rate Derived Core Temperature Algorithms for Use in NCAA Division I Football Athletes

**DOI:** 10.3390/jfmk5030046

**Published:** 2020-07-06

**Authors:** Joshua Hagen, Aaron Himmler, Joseph Clark, Jad Ramadan, Jason Stone, Jon Divine, Robert Mangine

**Affiliations:** 1Rockefeller Neuroscience Institute, West Virginia University, Morgantown, WV 26505, USA; joshua.hagen@hsc.wvu.edu (J.H.); jramadan@hsc.wvu.edu (J.R.); jason.stone1@hsc.wvu.edu (J.S.); 2Department of Athletics, University of Cincinnati, Cincinnati, OH 45221, USA; kuehnhav@ucmail.uc.edu (A.H.); joseph.clark@uc.edu (J.C.); divinej@ucmail.uc.edu (J.D.)

**Keywords:** exertional heat illness, athlete monitoring, core temperature

## Abstract

The purpose of this study was to assess the validity of utilizing heart rate to derive an estimate of core body temperature in American Football athletes. This was evaluated by combining commercially available Zephyr Bioharness devices, which includes an embedded estimated core temperature (ECT) algorithm, and an ingestible radio frequency core temperature pill during the highest heat injury risk timepoint of the season, summer training camp. Results showed a concordance of 0.643 and 78% of all data points fell within +/−1.0 °F. When the athletes were split into Upper (>/=6.0%) and Lower (<6.0%) body composition groups, there was a statistical improvement in accuracy with the Upper Body Fat% reaching 0.834 concordance and 93% of all values falling within +/−1.0 °F of the Gold Standard. Results suggest that heart rate derived core temperature assessments are a viable tool for heat stress monitoring in American football, but more work is required to improve on accuracy based on body composition.

## 1. Introduction

A growing abundance of applied sport science initiatives include strategies for mitigating the deleterious effects of heat stress, which are preventable through the implementation of proper training and effective heat stress monitoring (HSM) protocols. The ramifications of heat stress, such as heat-induced severe cramping, edema, rhabdomyolysis, or heat strokes to name a few, are often characterized as exertional heat illnesses (EHI) [1,2,3,4], and arise when homeostasis cannot maintain a core body temperature beneath 40.5 °C (considered as excessive elevations in core body temperature during physical exertion). This process is primarily achieved through constant heat exchange between the human body and the ambient environment via evaporation, radiation, convection, and conduction [2]. Athletes, whom inherently endure bouts of intense and voluminous physical exertion throughout their training cycles, rely on these thermoregulatory processes to counterbalance inevitable elevations in core body temperature as a result of metabolic heat production, a necessary by-product of exercise.

However, if an athlete’s physiological systems, namely their cardiovascular and nervous systems as well as their skin, are unable to prevent excess elevations in core body temperature then they are at a severe risk for injury, or in the most severe cases, fatality [2,5]. A position statement on exertional heat illnesses in collegiate athletes released in 2015 from the National Athletic Trainers Association related core temperatures exceeding 40.5 °C with exertional heat stroke [2], a catastrophic event for athletes that must be avoided at all costs. While not all heat illnesses are temperature-mediated, it is worth noting that exertional heat strokes are among the leading causes of sudden death in collegiate athletes [6]. Fortunately, heat stress morbidity and mortality are largely avoidable with proper HSM protocols.

In general, HSM strategies are designed to ensure the health and well-being of athletes, especially during training periods comprising bouts of high volume or intensity (often times training is a combination of both) conducted under extreme environmental conditions, such as high humidity and/or ambient temperatures. For example, American football is one (but not the only) sport that comprises high magnitudes of physical workload, often times performed in harsh environments [7,8,9]. Prior to each competitive season, football athletes participate in summer training camp, which imposes tremendous amounts of fatigue on the athletes and likely occurs during some of the hottest days within a calendar year [9,10,11]. In fact, the National Collegiate Athletic Association (NCAA) recently made drastic alterations to the practice specifications (e.g., number of practices per day, amount of collision contact) permitted during preseason training camp periods. The NCAA’s deliberate attempt to mitigate EHI is a result of the increased awareness surrounding EHI in athletes [1,2,6,12], which ultimately stem from tragic events that saturated mainstream media including but not limited to the deaths of pro-bowl offensive tackle, Korey Stringer, in August of 2001 and University of Maryland football athlete, Jordan McNair, who sustained an EHI in May 2018, and succumbed in June 2018. Both athletes suffered exertional heat strokes during the summer months. Moreover, the Coronavirus 2 (COVID-19) pandemic that halted the entire global sports domain in March 2020 and prevented training and monitoring protocols to progress normally poses a new and unique challenge to the sports performance realm as it is likely that athlete training statuses were negatively impacted [13]. Indeed, athletes are at increased risk for EHI when first returning to practice following a relatively long “break”, such as the initial summer months following the spring semester for collegiate fall athletes [3,4], which further exposes the importance of expeditiously incorporating HSM strategies into athlete training and recovery.

As previously alluded to, EHIs are largely preventable with careful considerations applied to HSM protocols that are implemented and refined throughout the various training periods (e.g., preseason, offseason, competitive season). Yet, typical procedures garnering research quality data that measure core body temperature comprise invasive and uncomfortable instruments, such as rectal or esophageal temperature probes, that are impractical for routine use in athletics. Thus, efforts to develop more practical approaches for measuring core temperatures ensued [14,15,16]. However, concerns regarding the validity and reliability for many of these instruments, such as axillary, oral, or tympanic sensors are questioned throughout the extant literature [16,17,18,19,20]. One alternative strategy utilizes telemetric pills that are swallowed by athletes and transported through their digestive system within 24–48 h [18,19,21]. Telemetric pills were previously validated as an acceptable instrument for quantifying core body temperature in humans [22,23], although limitations for daily or routine use beyond unit costs do exist. Unfortunately for many athletes, when telemetric pills are utilized, data collection typically requires the donning of a data logging device, which is impractical for contact sports such as American football, rugby, or mixed martial arts. Alternatively, remote transmissions of data are sometimes possible, which likely still necessitates that at least one human data recorder (presumably need much more for large groups of athletes such as American football teams) meticulously and manually records data throughout any given practice or training session. Furthermore, previous research reported that the accuracy of telemetric pills is sensitive to fluid ingestion [18] and ingestion timing [19], posing unique challenges for athletic groups that are constantly hydrating to prevent EHIs and other harmful effects of dehydration.

Interestingly, recent investigation of the utilization of serial raw heart rate signals to estimate core body temperature during ambulation reported an impressive overall bias of −0.03 ± 0.32 °C when compared to an ingested telemetric pill [14], which was later replicated in a similar model conducted in first responders [24]. The estimated core temperature (ECT) algorithm is based on a Kalman filter model using a series of heart rate (HR) measurements as the leading indicator; equations and MATLAB code available in Buller et al. [14]. The model contains two relationships, a time update and observational component. The ECT updates in 1-min increments by using updated HR data (time update) to modulate the previous value (observational). As such, the possibility for real time core temperature monitoring in athletes (applicable to military populations as well) through existing live athlete monitoring platforms that already obtain heart rate data garnered intrigue from researchers and practitioners alike [25,26,27]. Still, integrating the predictive algorithm developed through these previous efforts requires additional validations to ensure heart rate data obtained from the respective commercial devices are adequate enough for the algorithm to retain acceptable levels of accuracy. Further, while the published model for ECT [14,24] was rigorously developed and validated, this was done on non-athlete populations and not during sport-specific activities. Specifically, the nature of American Football in practice and game settings requires brief periods of high intensity effort followed by brief periods of rests (e.g., in-between plays) [7,28].

Therefore, the purpose of this study was to evaluate the accuracy of heart rate derived core body temperature estimations, which were obtained from a commercial electrocardiogram (ECG) chest strap using the published ECT algorithm. More specifically, data were collected on American football athletes during the highest risk time period of the season, summer training camp [3,4]. Temperature estimations were compared to a telemetric pill that was ingested by the athletes prior to several team practices. A secondary purpose was to examine potential influences that athlete body composition may impose on estimation accuracy. We hypothesized that ECT calculations would perform within 2 °F with respect to the telemetric pill. The +/−2 °F (1.1 °C) enables a range between 102.9 and 106.9 °F (39.4–41.6 °C, with a midpoint at 40.5 °C) thus allowing the monitoring of temperature ranges indicative of EHI risk [2]. Due to the novelty of this data collection, previous data were unavailable for use in determining a required sample size a priori. However, a post-hoc power analysis was conducted using Bland Altman statistics calculated from the data. Post-hoc power analysis will aid in justifying results presented herein and will be considered for future applications of ECT in sport and research. These data will reveal critical information relevant to athletes and sport practitioners interested in HSM as a strategy for mitigating EHI.

## 2. Materials and Methods

### 2.1. Participants

A total of 13 male Division I football athletes participated in this study. Demographic information is listed in Table 1. All subjects gave their informed consent for inclusion before they participated in the study. The study was conducted in accordance with the Declaration of Helsinki. This research was covered under the approved IRB Protocol at the University of Cincinnati (#2017-3008, Approval #00003152). The athletes were additionally grouped into “Upper BF%” (>6.0 Body Fat) and “Lower BF%” (</=6.0% Body Fat) to further understand any links between accuracy and body composition. Positions included in the split are denoted in Table 1, which comprised Offensive Tackle (OT), Line Backer (LB), Tight End (TE), Running Back (RB), Safety (S), Cornerback (CB), Wide Receiver (WR), and Quarterback (QB). Subjects were eligible to volunteer for this study first by being members of the University of Cincinnati Football team and full participants in practice (assured medical clearance). Exclusion criteria included athletes who were not cleared for full participation in practice.

### 2.2. Technologies

Two technologies were utilized for assessing core body temperature. The first was the reference value provided by ingestible core temperature pill, which was the CorTemp Ingestible Core Body Temperature Sensor manufactured by HQInc [29]. The sensor was encased in a Silicone Coating, and consisted of a battery, circuit board, quartz crystal, and Radio Frequency coils. The quartz crystal oscillated at a frequency which is temperature dependent, which then transduced the signal to a readable core temperature value. The sensor passes through the body at a normal motility rate of 24–36 h, and has been assessed as accurate to +/−0.1 °C. Additionally, this device is FDA cleared as a single-use device. The data was displayed using the Wireless Core Body Temperature Monitoring Data Recorder operated in “Sports Mode”. The second technology utilized was the Zephyr Bioharness 3 from Medtronic [30]. The Bioharness 3 was a single lead 1000 Hz ECG based sensor for measuring heart rate, heart rate variability, and movement via accelerometry. The Bioharness 3 puck was attached to the Zephyr chest strap, and then connected wirelessly to OmniSense Live (Version 5.0.0) where live data was displayed on a Windows PC. Raw data was available post measurement via USB download to OmniSense Analysis (Version 5.0.0) software. Included in both the Live and Downloaded data was the “Estimated Core Temperature” algorithm [14], which was directly compared to the Core Temperature Pill Sensor.

### 2.3. Process

Three hours prior to practice, the Core Temperature Pill Sensor was prepared by removing from the sealed package and removing the plastic strip that activated the battery in the sensor. The sensor was then ingested by the participant, and sensor number was associated with the participant’s jersey number to ensure the correct association. The 3-h pre-practice time window ensured that the pill was settled in the stomach cavity during practice and would have minimal impact from ingesting water on the temperature reading. This 3-h window was selected based on (1) supporting literature [18] showing in some cases that 3 h was sufficient time to eliminate interference from cold water ingestion and (2) logistics of implementing the pill with early practice times while not risking the loss of a pill from fast motility. Less than 30 min before the start of practice, each participant was given a Zephyr Bioharness 3 puck attached to a Zephyr strap that was again assigned according to their jersey number for proper association. The ECG leads were wetted to promote a strong signal from the start of wearing. When the practice started, the OmniSense Live software streamed live Heart Rate and Estimated Core Temperature, termed “Zephyr ECT”, data to a PC wirelessly on the sideline. The “Sports Mode” of the Core Body Temperature Data Recorder allowed for the connection to multiple Core Sensors, but must be placed by the lower back of the athlete to transmit a single data point, which for this study was termed the “Gold Standard”. During breaks between practice sessions (between 10 and 20 breaks commonly), personnel would run to the participant with the Data Recorder, a notebook, and the OmniSense Live computer. Data was manually recorded for (1) Timestamp, (2) Jersey Number, (3) Core Pill Temperature (Gold Standard), (4) Zephyr ECT. Post practice, this data was transferred to Microsoft Excel, and the Zephyr Bioharness 3 pucks were connected to the PC to offload the raw data from the session.

### 2.4. Data Processing

After the Zephyr Bioharness data was downloaded and processed in OmniSense Analysis, the Zephyr ECT values were verified versus the timestamps and jersey numbers to ensure the proper data was recorded on the field. Further visual analysis was done for each participant’s data to ensure a Heart Rate Confidence value of 80% or higher, which verifies a clean ECG signal and good data. Any data with Heart Rate Confidence issues were rejected from analysis. Data analysis was performed using Microsoft Excel and IBM SPSS, with more details in Section 3.

### 2.5. Statistics

The primary interest was comparing values from Zephyr ECT to the Gold Standard ingestible pill. Thus, in this experiment, an observation from this dataset was in the form of a bivariate matched pair. Data was investigated in three ways, using Lin’s concordance correlation coefficient (CCC) [31], Bland–Altman analysis [32], and histograms that look at the distribution of differences in measurement between the Zephyr ECT and Gold Standard. Lin’s CCC is an established analysis for evaluating agreement [2] in assessing variation from a single metric to a gold standard reference, and has been used previously in validating heart rate measurements [3]. Bland–Altman analysis produces visualization of bias along with limits of agreement, and has been widely used in previous research assessing validation of physiological devices [3,5,6,7,8,9,10].

The CCC measured the strength of the bivariate relationship between the Zephyr ECT and Gold Standard in comparison to the identity (y = x) line. The identity line should be used in this case, as it represents the situation of perfect agreement between the two devices. The CCC, just like the traditional Pearson correlation coefficient, has a range of ±1. The closer the value to 1, the stronger the level of agreement between the two devices. Plots are also included to help visualize this concept.

For a Bland–Altman analysis, the difference between the matched pair was plotted against the mean of the two matched pair measurements. The difference was calculated as:Difference (°F) = Zephyr ECT (°F) − Gold Standard (°F)(1)

The average of the differences was interpreted as the bias, (used to understand whether Zephyr ECT overestimates or underestimates the core temperature), while the standard deviation of the differences was used to calculate the lower and upper and limits of agreement (bias ± 1.96 * SD). These three values were added as horizontal reference lines on the Bland–Altman plot to provide a good visualization of overall device measurement performance. The values of the bias and limits of agreement, as well as their associated 95% confidence intervals, will be reported in a table.

### 2.6. Environmental Conditions

Testing occurred during Summer Training Camp for Division I Football in August in Southwest Ohio and Southeast Indiana. Information about practice start times and environmental temperature is listed in Table 2 [33].

### 2.7. Training Load Description

All of the data collection occurred during summer camp Football practices, all of which were scripted similarly. Practices had between 18 and 22 periods lasting 5 min each. Each practice began with a teaching or walk through period which were led by position coaches, followed by a specialist period, then followed by a team dynamic warmup and stretching period. Then an opening technique period with position specific drills, again followed by a specialist period. From here, a bulk of the practice consisted of 12–16 periods, which can vary between individual position sessions to mini field to full field run throughs. The nature of American Football practice and game demands require many brief periods of high intensity efforts, followed by brief periods of rest until the next play or drill starts [7].

## 3. Results

### 3.1. Analysis of All Participants

A total of 134 discrete measurements were collected across all participants and practices. A single measurement was defined as a value provided by the ingestible pill (Gold Standard) read locally to an RF reader, and recorded along with a timestamp to the nearest second. The Zephyr ECT value was manually recorded at the same time via a telemetry system on a laptop PC, and further validated against downloaded and processed data post-practice. The RF reader value corresponded to the “Gold Standard” value in equation 1, and Zephyr ECT value checked against the downloaded data corresponds to “Zephyr ECT”. The Bland–Altman plot in Figure 1 gives an overall picture of Zephyr ECT measurement error for all participants across all practices.

The vast majority of measurements are within ±2 °F. Table 3 below gives more details on the Bland–Altman statistics for comparison of Zephyr ECT to Gold Standard.

Overall, the bias value was −0.192, which means that Zephyr ECT missed the Gold Standard value by, on average −0.192 °F. The 95% CI for the bias value was (−0.333, −0.05). Since “0” is not contained in this interval, then at the 0.05 level, it can be concluded that Zephyr ECT consistently underestimated the Gold Standard temperature, although minimally.

The LOA range, (−1.803, 1.42), represents the range where 95% of the differences between Zephyr ECT and Gold Standard are expected to fall. These limits have confidence intervals of their own. Looking at the values, it is not out of the realm of expectations for a Zephyr ECT measurement to underestimate the Gold Standard by 2 °F.

Next, the concordance correlation coefficient (CCC) provides a single summary measure of the strength of the relationship between Zephyr ECT and Gold Standard measurements. Figure 2 below provides a visualization of this.

The green dashed lines in the plot 0.5 °F above and below the concordance line represent “very good” agreement. The CCC (along with associated 95% confidence intervals) is listed in Table 4 below.

The actual concordance value may not have much meaning in this context, as there is no other group or dataset being used in comparison (however this will be useful later to compare subgroups).

Now, the focus turns to the distribution of the differences between Zephyr ECT and Gold Standard measurements. The histogram in Figure 3 shows that 78% of the data points fell within +/-1.0 °F, and 54% within +/−0.5 °F for all participants across all practices.

### 3.2. Analysis of Position Grouping

As stated in the methods, observations were further grouped by “Upper Body Fat% (BF)” and “Lower Body Fat% (BF)”, and it was of interest to investigate any differences in measurement accuracy between the two groups. First, a side-by-side Bland–Altman comparison shown in Figure 4.

The “Upper BF%” plot clearly showed a much smaller spread in difference between temperature readings compared to “Lower BF%”. Table 5 lists the Bland–Altman statistics of these two plots.

With regard to the bias, the players with higher BF% were seen to have their temperatures consistently underestimated by Zephyr ECT, as the 95% CI did not include “0”. This underestimation was minimal at −0.247 °F. The 95% CI for bias for players in the Lower BF% group did not present enough evidence to support this claim thus a case can be made that Zephyr ECT measurements on Lower BF% players are unbiased. For players in the Upper BF% group, the range between upper and lower LOA was calculated to be 0.766 – (−1.26) = 2.026, while for the Lower BF% group, the range was considerably larger at 1.77 – (−2.076) = 3.846. Thus, Upper BF% participants exhibited much more precision than Lower BF% in their Zephyr ECT measurements.

One way to compare the overall performance between “Upper BF%” and “Lower BF%” groups in accuracy compared to Gold Standard temperature is the CCC. Differences from the concordance line within subgroup are shown in Figure 5 below.

Similar to the Bland Altman analysis, Lower BF% participants were seen to have a higher spread around the concordance line compared to Upper BF%. Next, CCC values are presented in Table 6.

Upper BF% participants showed a higher CCC value compared to Lower BF%. The confidence intervals between the two groups do not overlap thus, at the 0.05 level, there is enough statistical evidence to conclude that the level of agreement (concordance) in temperature measurements was significantly higher in Upper BF% participants when using Zephyr ECT to measure core body temperature.

Lastly, when looking at the distribution of the differences in temperature measurements in Figure 6 and Table 7, it is clear that a much higher percentage of Zephyr ECT values in the Upper BF% group fell within 1.0 °F.

Due to the applied nature of this study, which is a common byproduct of studying sport, a post hoc power analysis was completed. With an acceptable range set at +/−2 °F, we set the lower limit of agreement “L” as −2 °F and upper limit of agreement “U” as 2 °F. If the lower limit of the 95% CI for L, and the upper limit of the 95% CI for U fall within (−2, 2), there is enough evidence to conclude the two methods “agree” as per the agreement requirement. Thus, we tested two hypotheses at the same time, and both need rejecting:

**Hypothesis** **1** **(H_01_).**
*L < −2; H_A1:_ L ≥ −2.*


**Hypothesis** **2** **(H_02_).**
*U > 2; H_A2:_ U ≤ 2.*


To power this, two values were used [34]: standardized difference limit (the mean of differences divided by the standard deviation of the differences), and the standardized agreement limit (2 divided the standard deviation of the differences). Here, the standardized difference limit is −0.192/0.822 = −0.23, and the standardized agreement limit is 2/0.822 = 2.43. With these two values, a sample size of 402 measurement pairs would be required to achieve 80% power at the 0.05 level for the hypothesis test above. However, when this same post hoc power analysis was run using only the Upper BF% group data, only 14 measurement pairs would be required to achieve 80% power at the 0.05 level.

## 4. Discussion

The primary motivation behind this study was to assess the accuracy of using heart rate data to derive an estimated core body temperature in Division I NCAA football athletes during the highest EHI risk time of the season. When assessed for every participant across all practices, the result was a CCC of 0.643. Statistically with CCC, this falls in the “poor” range, but this translates to 78% of the values falling between +/−1.0 °F. The decision to utilize this technology and method during training comes down to the practitioners, and if they are willing to accept that most data points are within +/− 1.0 °F accuracy to help augment their HSM surveillance process.

More interestingly, when the participants were split by body composition into an Upper BF% group and Lower BF% group, the results were very different. The Upper BF% group, defined as above 6% BF with a range between 6% and 18.8%, showed a substantially and statistically significant improvement in CCC with a value of 0.834, compared to 0.515 for the Lower BF% group. This is also reflected in an improvement in percentage of values within +/−1.0 °F accuracy up to 93%, as compared to 62% for the Lower BF% group.

The ECT algorithm has been well published, cited, and used in military populations, but not formally investigated for validity in athletics. Currently, this algorithm is only specific to heart rate data, and the model was trained and validated on military personnel. One explanation for the increased accuracy for the Upper BF% group is that the ECT algorithm was trained on participants that were in a higher body composition range, of 13–18% [14], which closer matches the Upper BF% group compared to the Lower BF% group. Physiologically, there are at least two other factors to consider due to body composition differences, surface area and metabolic demand of tissue. Potential differences in athletes’ body mass-to-body surface area ratio (BM/BSA) might suggest that there may be a greater surface area available for evaporative cooling based on upon an single athlete’s anthropometric/body composition makeup [35]. Therefore, future developments on ECT algorithms should take this into account. A second factor to consider comprises the differing metabolic demands of skeletal muscle tissue compared to adipose tissue. Muscle tissue possesses inherently greater metabolic demands, both at rest and during exertion, when compared to fat thus impacting the overall energy expenditure, which should be investigated as an additional factor for thermal regulation applications. Future studies powered to and designed to incorporate body composition and BSA are being considered.

Due to the applied research aspect of this study, several limitations are evident. (1) Number of subjects: due to the cost associated with the one time use core temperature pills, a certain number were able to be procured with the budget provided. This number was distributed across 13 athletes to enable multiple training sessions with the same subjects. A follow-on study should include a larger number of athletes and use the data collected in this study to provide power analysis to include proper distribution among different body compositions for investigating updated algorithms. (2) Number of data points: the pills selected for this study were not able to log data, and required a reader to be placed on the back of the subject for data transfer. This significantly limited the total number of data points, with trainers only able to collect data between series during training. Future research should utilize a pill that enables continuous logging at 30–60 s increments that allows for post-training download. This will enable a single subject to collect between 120 and 240 unique data points during a 2-h practice, and will be well above the full group post hoc power analysis within a handful of subjects. (3) Potential interference with cold water ingestion: due to the early morning practice times, a 3-h pre-training ingestion time was logistically feasible, whereas a 6–8 h window that likely could further mitigate interference would be more ideal. (4) Based on the assessment of the core pills, a majority of the core temperatures in the subjects did not exceed 102.5 °F. Additional data should be collected to investigate data points exceeding 105 °F where clinical significance becomes more evident. By expanding the number of subjects and data points suggested previously, the potential for expanding the range of temperatures is likely to increase.

This, along with the data presented in this study suggest that modifications can be investigated in the ECT algorithm to improve accuracy for lower body composition athletes. Due to the nature of American Football, additional investigation for football specific exertion should be studied. This is a sport with substantial equipment worn, and repeated high intensity bouts with rest in-between. This can be 10–20 s between plays, and 5–10 min between series during practices and games. This differs greatly from sports like soccer, where long durations of aerobic and anaerobic work is endured. Even more different are tactical populations like the military which can endure hours upon hours of steady state movements with substantial equipment. It is possible that enhanced algorithms can be created for these cases.

## 5. Conclusions

HSM is an important tool that can be used in exertional training settings to help reduce the risk of EHIs. However, for this to become standard practice, scientifically validated tools must be used, but just as important, must be logistically feasible to implement on a daily basis by the training staff. Data must be trusted, reliable, and instantly actionable. The data presented in this study shows that deriving ECT off of accurate heart rate data shows promise in HSM applications. Additionally, the ECT algorithm was implemented in a commercial monitoring system and can be added to other systems with minimal software modifications, which is important for daily use applications and not research. It is ultimately up to the practitioners what level of error they are comfortable with in using this as a surveillance tool, but this data shows that it is feasible and logistically possible with commercial systems.

Future work should be done investigating modified ECT algorithms off of heart rate that are specific to body composition and sport/position for athletics applications. For tactical and military populations, additional algorithm work could include age, body composition, gender, specialty. Additionally, as the wearable technology sector continues to evolve, additional sensors should be investigated for inclusion in algorithm work as well, such as accelerometers, skin temperature, ambient temperature, and potentially sweat rate.

## Figures and Tables

**Figure 1 jfmk-05-00046-f001:**
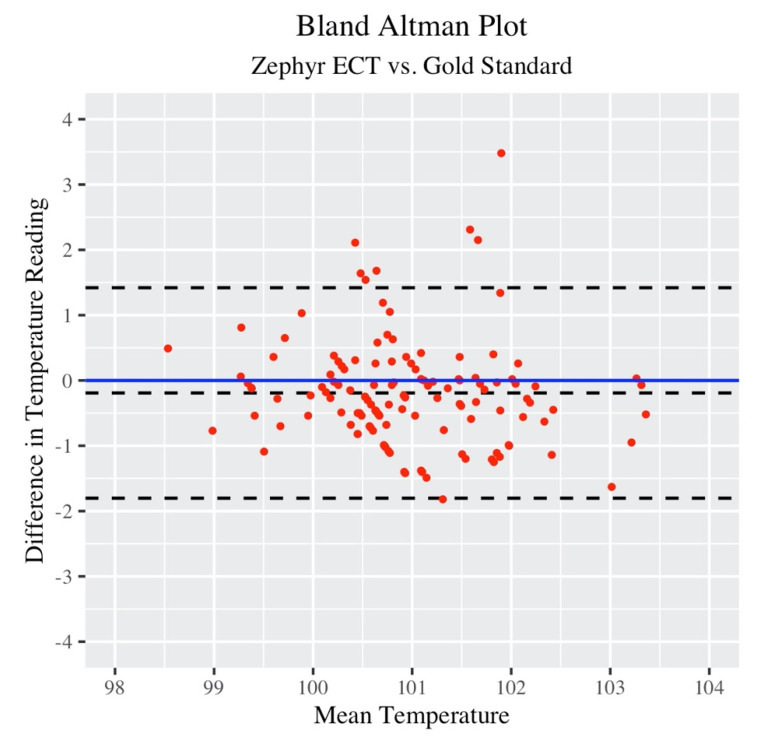
Bland–Altman plot: Zephyr estimated core temperature (ECT) versus Gold Standard.

**Figure 2 jfmk-05-00046-f002:**
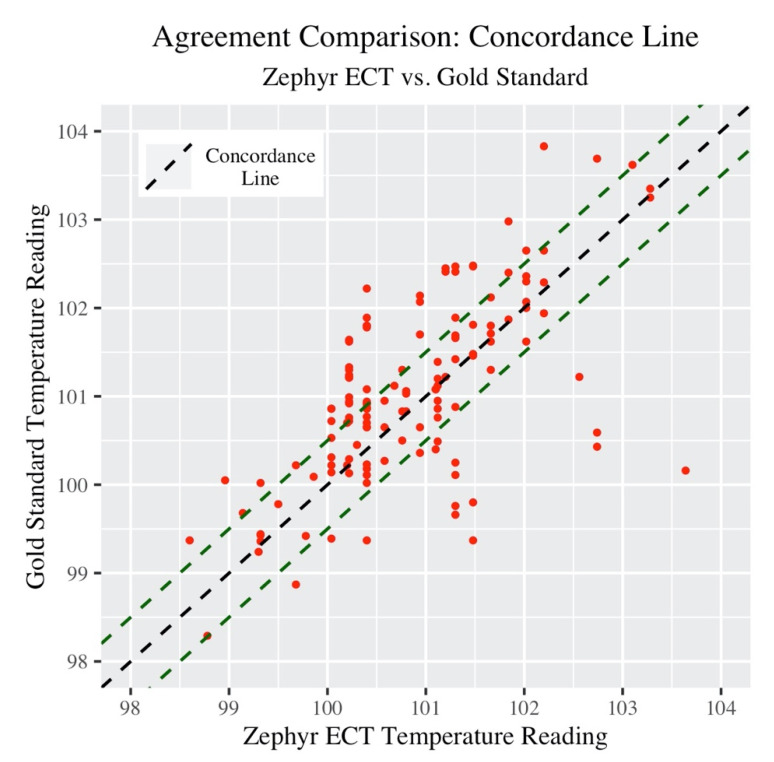
Concordance plot: ECT versus Gold Standard.

**Figure 3 jfmk-05-00046-f003:**
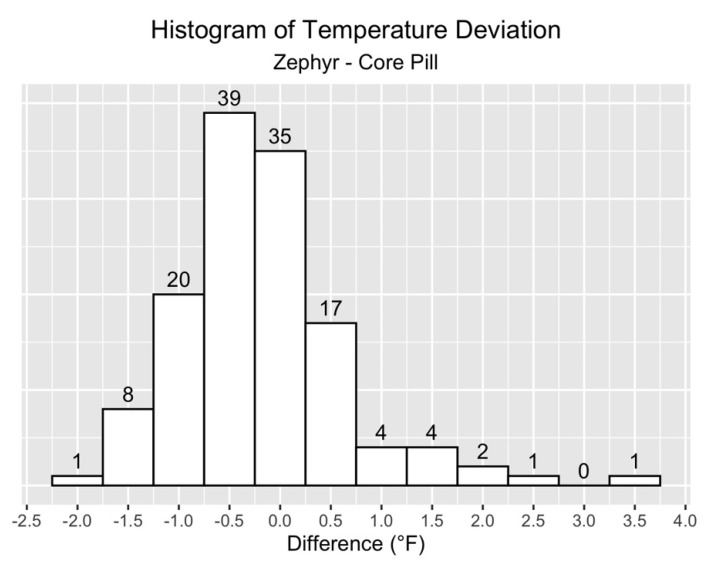
Histogram of ECT versus Gold Standard: number of observations (Y-axis) within given ranges of error (X-axis).

**Figure 4 jfmk-05-00046-f004:**
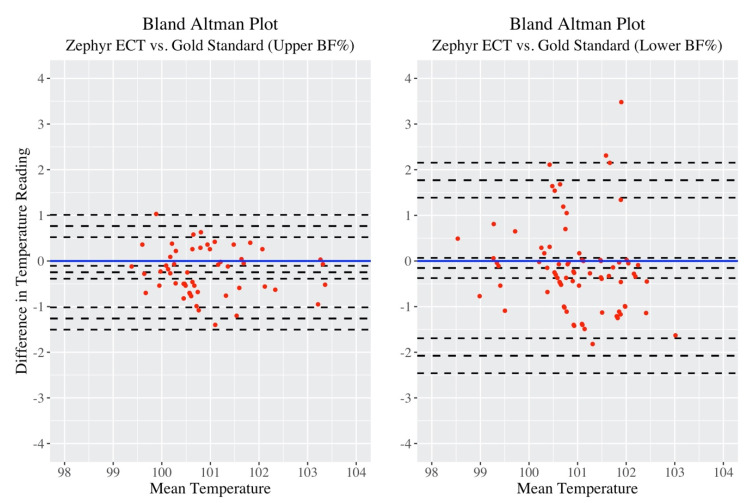
Side-by-side Bland–Altman plots for Upper BF% (left) group vs. Lower BF% (right) group.

**Figure 5 jfmk-05-00046-f005:**
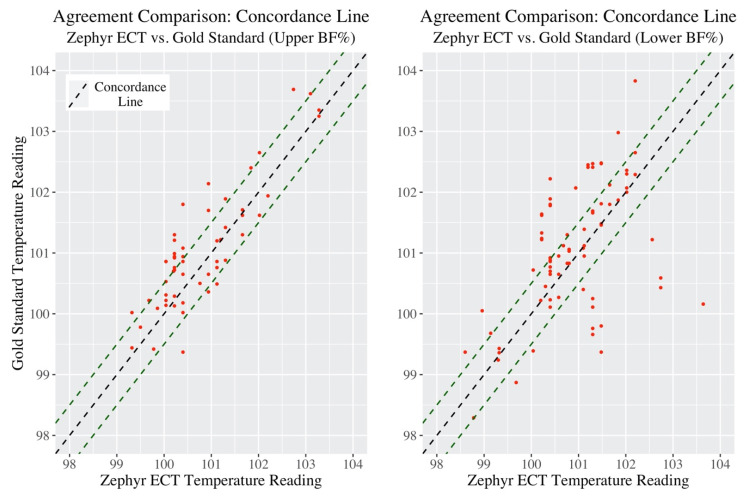
Concordance plots of Upper BF% group (left) and Lower BF% group (right).

**Figure 6 jfmk-05-00046-f006:**
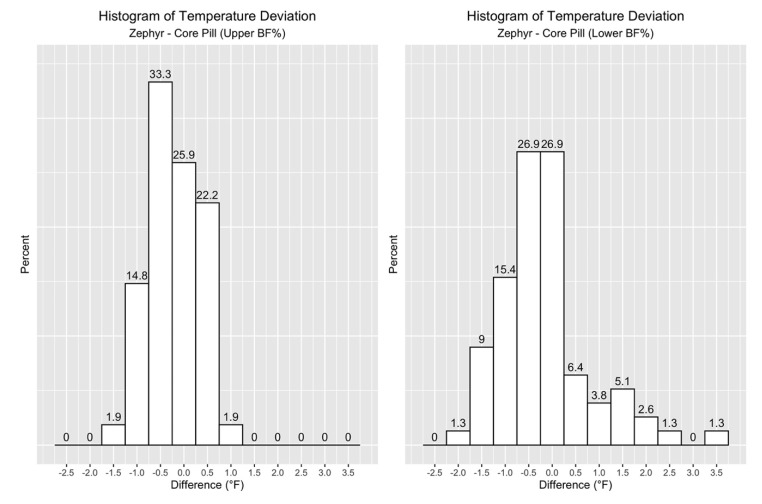
Histogram plots for Upper BF% (left) and Lower BF% (right) groups, where the percent of total observations (Y-axis) is plotted against given ranges of error (X-axis).

**Table 1 jfmk-05-00046-t001:** Characteristics of participants.

Characteristic	All Participants	“Upper BF%”	“Lower BF%”
Positions included		OT, LB, TE, RB, S	S, CB, WR, QB, LB
Number of participants	13	6	7
Height range (in)	70–78”	70–78”	70–75”
Height average (in)	73.2”	74.2”	72.4”
Weight range (lbs)	178–315	212–315	178–215
Weight average (lbs)	217	240	196
Body fat range (%)	3–18.8	6.7–18.8	3–6.0
Body fat average (%)	7.3	10.1	4.8

**Table 2 jfmk-05-00046-t002:** Practice conditions.

Practice Number	Practice Start Time	Practice Dry Bulb Temperature (°F) (Start/Finish)	Practice Wet Bulb Temperature (°F) (Start/Finish)	Practice Relative Humidity (%) (Start/Finish)
1	7:00 pm	81/69	68/66	51/87
2	11:00 am	81/85	71/70	61/46
3	9:00 am	80/85	72/74	79/61
4	9:00 am	81/88	76/79	79/66
5	12:00 pm	76/78	62/63	43/43
6	1:00 pm	79/81	68/68	54/49

**Table 3 jfmk-05-00046-t003:** Bland–Altman statistics for Zephyr ECT vs. Gold Standard.

Bland-Altman Statistic	Value	Lower 95% CI	Upper 95% CI
Bias (Δ°F)	−0.192	−0.333	−0.05
Lower LOA	−1.803	−2.046	−1.56
Upper LOA	1.42	1.178	1.662

**Table 4 jfmk-05-00046-t004:** Bland–Altman statistics for Zephyr ECT vs. Gold Standard.

Statistic	Value	Lower 95% CI	Upper 95% CI
CCC	0.643	0.534	0.731

**Table 5 jfmk-05-00046-t005:** Subgroup Bland–Altman statistics: Upper BF% and Lower BF%.

Group	Bland–Altman Statistic	Value	Lower 95% CI	Upper 95% CI
Upper BF%	Bias (**Δ**°F)	−0.247	−0.388	−0.106
Lower BF%	Bias (**Δ**°F)	−0.153	−0.374	0.068
Upper BF%	Lower LOA	−1.26	−1.502	−1.017
Lower BF%	Lower LOA	−2.076	−2.456	−1.697
Upper BF%	Upper LOA	0.766	0.524	1.009
Lower BF%	Upper LOA	1.77	1.39	2.15

**Table 6 jfmk-05-00046-t006:** Subgroup concordance correlation coefficients (with 95% CI).

Group	Statistic	Value	Lower 95% CI	Upper 95% CI
All Participants	CCC	0.643	0.534	0.731
Upper BF%	CCC	0.834	0.734	0.898
Lower BF%	CCC	0.515	0.336	0.659

**Table 7 jfmk-05-00046-t007:** Histogram values for Figure 6.

Group	Values within +/−0.5 °F	Values within +/−1.0 °F
All	54%	78%
Upper BF%	57%	93%
Lower BF%	46%	62%

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
