# Peer review of "Test and Evaluation of Heart Rate Derived Core Temperature Algorithms for Use in NCAA Division I Football Athletes"

_jfmk, 2020, doi:10.3390/jfmk5030046_

Round 1

Reviewer 1 Report

The authors are to be commended for collecting important data in a field setting.  I have some suggestions that may improve the final product.

1) Page 1- Line 28: Also and very importantly, rhabdo.  

2) Page 1- Line 30: maintain temp near 37 degrees C?  Not during exercise.  

3) Page 1- 1st paragraph: some of the conditions listed are likely not temperature mediated.  

4) Page 2- Line 49: Fall training camp?  Do you mean summer camp?

5) Page 2- Line 57: Jordan suffered his EHS in May 2018 and succumbed to the complication in June 2018.  

6) Page 3- Line 123: I am not aware of any methodological studies with ingestible thermistor that suggest 3 hours as the procedure to follow for ingestion prior to use.  I think 5-6 hours would be better.  Please provide data from references to substantiate this decision. 

7) Page 5- Table 2: please provide WBGT and sub-component parts in this table.  Ambient temperature alone has very little utility when examining how enviornment is influencing the validity of the device.  

8) Methods- Please provide great detail regarding the practice sessions in which data was collected.  The prediction may have better predictive value during steady-state exercise versus exercise that is dynamic and intensity makes massive swings in short time periods.  

9) All but 5 readings under 103 and about 3/4 under 102.  Need to be clear in discussion about limited clinical applications since most football players would be in 102.5 to 105 range during intense exercise session in the heat.  

10) ECT algorithm- please explain how this calculated.  

11) Discussion- please provide great detail about limitations of the study.

Author Response

Thank you for your thoughtful comments.  They have all been individually addressed with changes reflected in the attached updated manuscript, and documented with line numbers below:

1) Page 1- Line 28: Also and very importantly, rhabdo.  Added to line 28

2) Page 1- Line 30: maintain temp near 37 degrees C?  Not during exercise.  Line 30:  Clarified to replace 37 degrees C with preventing excess temperature rises above upper limits associated with EHI (40.5)

3) Page 1- 1st paragraph: some of the conditions listed are likely not temperature mediated.  Added in line 42

4) Page 2- Line 49: Fall training camp?  Do you mean summer camp?  Modified – now in Line 51

5) Page 2- Line 57: Jordan suffered his EHS in May 2018 and succumbed to the complication in June 2018.  Modified in line 59-60

6) Page 3- Line 123: I am not aware of any methodological studies with ingestible thermistor that suggest 3 hours as the procedure to follow for ingestion prior to use.  I think 5-6 hours would be better.  Please provide data from references to substantiate this decision.  Added a citation and rationale for selecting this time window in line 160, and added to the Limitations section starting in line 424.

7) Page 5- Table 2: please provide WBGT and sub-component parts in this table.  Ambient temperature alone has very little utility when examining how enviornment is influencing the validity of the device.  Added DBGT, WBGT, RH to Table 2 in line 217

8) Methods- Please provide great detail regarding the practice sessions in which data was collected.  The prediction may have better predictive value during steady-state exercise versus exercise that is dynamic and intensity makes massive swings in short time periods.  Added to Methods section on line 220-228, and statement in line 103 in introduction

9) All but 5 readings under 103 and about 3/4 under 102.  Need to be clear in discussion about limited clinical applications since most football players would be in 102.5 to 105 range during intense exercise session in the heat.  Added to limitations section in line 426

10) ECT algorithm- please explain how this calculated.  Added to line 92

11) Discussion- please provide great detail about limitations of the study.  Study limitations section added at line 412 in the Discussion section

Reviewer 2 Report

The purpose of this study was to assess the validity of utilizing heart rate to derive an estimate of core body temperature in American Football athletes. The authors concluded that heart rate derived core temperature assessments are a viable tool for heat stress monitoring in American football.

The study is important; but it present some limitation, especially the limited number of participants, and some modifications and/or explanations are needed.

Introduction

I suggest adding clear hypothesis based on the analysis of the literature.

Methods

I suggest presenting the participants at the beginning of this section and to add the calculation and justification of the sample size.

Also, I suggest including the inclusion and exclusion criteria of participants.

Why the authors have used Lin’s Concordance Correlation Coefficient (CCC) and Bland-Altman analysis?

Please provide the approval number of the Ethical committee approval?

Discussion

A paragraph stating the limitations of the study should be added at the end of the discussion.

Conclusion

I suggest adding clear practical recommendations

Author Response

Thank you for your thoughtful comments.  All have been addressed individually with line numbers below, and captured in the attached revised manuscript.

Introduction

I suggest adding clear hypothesis based on the analysis of the literature.  Added to line 112

Methods

I suggest presenting the participants at the beginning of this section and to add the calculation and justification of the sample size.  Moved Participants to beginning of Methods in line 122.  Added information in Introduction section on line 115.  Post Hoc power analysis reported starting at line 357

Also, I suggest including the inclusion and exclusion criteria of participants.  Added to line 130

Why the authors have used Lin’s Concordance Correlation Coefficient (CCC) and Bland-Altman analysis?  Added in lines 188-192

Please provide the approval number of the Ethical committee approval?  Added to line 126

Discussion

A paragraph stating the limitations of the study should be added at the end of the discussion.  Limitations section added starting at line 412 in the Discussion section

Conclusion

I suggest adding clear practical recommendations.  Added into the discussion within the Limitations section starting on line 412

Round 2

Reviewer 1 Report

Excellent modifications.  

Reviewer 2 Report

The authors have properly responded to my first comments and this version is suitable for publication.